# Polishing Characteristics of Cemented Carbide Using Cubic Boron Nitride Magnetic Abrasive Powders

**DOI:** 10.3390/mi13122167

**Published:** 2022-12-08

**Authors:** Pengfei Chen, Yuewu Gao, Yugang Zhao, Guoyong Zhao, Guixiang Zhang, Haiyun Zhang, Zhuang Song

**Affiliations:** Institute for Advanced Manufacturing, Shandong University of Technology, Zibo 255049, China

**Keywords:** magnetic abrasive finishing, magnetic abrasive powder, cemented carbide, surface roughness

## Abstract

This paper describes the application of bonded magnetic abrasive powders (MAPs) in the magnetic abrasive finishing (MAF) process. In order to improve the poor finishing performance and short service life of MAPs in polishing super-hard materials, a double-stage atomization technique was used to successfully manufacture MAPs with a CBN as an abrasive phase. The prepared results show that CBN abrasives with their original structure were deeply and densely embedded on the surface of spherical MAPs. Based on the MAF process, a five-level and four-factor central composite design experiment was carried out to verify the developed MAPs polishing performance on the finishing of cemented carbide parts (864 Hv). Working gap, rotational speed, feed rate of a workpiece, and mesh number of MAP were considered as influence factors. The analysis data was used to understand different interactions of significant parameters. A regression model for predicting the change of surface roughness was obtained, and the optimal parameter combination was figured out through a solution of a quadratic equation in Design-Expert software. According to MAF results, the strong cutting ability of atomized CBN MAPs improved the surface roughness of cemented carbide by over 80% at the optimum parameters. The strong cutting ability of atomized CBN MAPs can produce good surface quality on the hard materials. The findings of this research can promote a large-scale application of MAF technology in the surface polishing of hard materials.

## 1. Introduction

Micro-and nano-level surface finishes of hard materials are gaining importance in this era. Cemented carbide is a type of hard material with high hardness, wear resistance, strength and corrosion resistance. It is suitable for manufacturing wear-resistant parts and cutting tools. The surface finish of a material will affect its functional properties. Therefore, many advanced manufacturing technologies have been developed to polish the workpiece surface, such as magnetorheological fluid-based finishing (MRFF) [1,2], magnetic abrasive finishing (MAF) [3,4,5], elastic abrasive finishing [5] and abrasive flow machining (AFM) [6,7]. In these methods, the movement of tiny ceramic particles (Al_2_O_3_/SiC) can remove the burrs from the surface. MAF is an important mode of abrasive flow finishing applied to a polished workpiece surface; it causes little surface damage and is convenient to control [8].

Many researchers have studied the finishing of different workpieces by using of a mixture of ferrous particles and abrasives (unbonded MAPs) in the MAF process. Kala and Pandey developed a double magnetic pole for polishing two different hardnesses of paramagnetic material [9]. They reported the rotational speed was the most significant influence parameter for improving the surface roughness of soft material. For hard stainless steel materials, lower working gap, which caused a better surface roughness. Furthermore, they studied the normal torque and finishing torque of unbonded MAPs in the MAF process [10]. They obtained the lower working gap had a great effect on both torque. Verma et al. [11] had evaluated the MAF process on the internal surface of stainless steel pipe. They developed a lower and upper permanent pole to perform the experiments. They obtained a surface roughness of 56 nm and reported that the working gap was the most important factor for surface finish. Kajal et al. [12] also investigated the surface roughness mechanism of the MAF process on the internal surface of the revolver barrel. They developed a theoretical model and verified the validity of the model through a full-factorial experiment. They reported that the helical grooves affected the polishing process of the magnetic abrasive brush.

Some studies have been reported the polishing quality of hard material in the MAF process. Mulik and Pandey [13] polished the AISI 52100 steel (61 HRC) by using unbonded SiC MAPs in the MAF. They obtained a fine surface finish with a roughness of 51 nm in a short processing time. In another study [14], the same work material was polished by ultrasonic-assisted MAF process. They compared the performance of two processing ways and obtained a better surface roughness in the UAMAF process. According to the above literature survey, although unbonded MAPs can successfully finish many different workpieces, the little holding capability of a formed ferromagnetic chain may affect the cutting force of hard particles.

Therefore, bonded MAPs were prepared via different preparation methods and had been used in the MAF process [15,16,17]. Lin et al. [15] polished the free-form surface of SUS304 material using sintered alumina MAPs. They used a Taguchi design to study the effect of different process parameters on the surface roughness of SUS304. They reported that sintered alumina MAPs provided a mirror-like surface. Wang et al. [16] used silicone gel to develop a bonded SiC MAP and finished the cylindrical rod of mold steel. They reduced the surface roughness of the part to 38 nm in 30 min. They observed that the finishing efficiency of bonded SiC MAP was three times that of unbonded MAPs. Hanada and Yamaguchi [17] produced the bonded MAP via a plasma spraying method to solve the problem of poor wettability between ferrous particles and abrasives. The prepared alumina MAPs have good sphericity and excellent finishing performance on the SUS304 stainless steel plate.

In previous studies, we had successfully prepared spherical alumina bonded MAPs [18] via an atomization technique and the polishing performance of developed MAPs on the finishing of copper alloy, stainless steel [19] and Inconel718 [20] was excellent. The characteristics of atomized bonded MAPs, such as uniform cutting depth of hard particle, sufficient cutting force and dense arrangement of spherical MAPs easily produced a smooth surface on the workpiece. However, in our following experiments, the use of alumina bonded MAPs produced a poor surface roughness on the hard materials, such as ceramic and cemented carbide. According to SEM images of the finished workpiece, some original burrs and grinding marks are still present on the surface. The weak wear resistance and hardness of abrasive make developed MAPs perform poorly in the material removal efficiency.

In order to improve the finishing performance of MAP, a super-hard cubic boron nitride (CBN) fine particle was selected as the abrasive. CBN abrasives and metal liquid were atomized by a double-stage atomization. A MAF test was performed to evaluate the polishing performance of atomized MAPs on cemented carbide. Furthermore, a central composite design technique was carried out to understand the finishing characteristics of CBN MAPs. The working gap between the magnetic pole and the workpiece, the rotational speed of the magnetic pole, the feed rate of the workpiece and the mesh number of MAPs four process parameters were selected as influence factors. The surface morphology of finished workpiece was obtained by scanning electron microscopy and used to verify the effect of process parameters on the surface roughness.

## 2. Experiment Procedures

### 2.1. Preparation of MAP

In this study, CBN particle (d_50_ = 14 μm) was chosen as the abrasive phase of MAP. An excellent magnetic alloy with the composition of Fe–3.7Si–1Ni–1Cu (wt.%) was used as the iron matrix of MAP. Figure 1 shows the schematic diagram of MAPs prepared by the double-stage atomization. The self-designed hard particle feeder can form a uniform N_2_-CBN two-phase flow. Open valves 1 and 2, and the N_2_ will fill with the particle tank and disperse CBN particles. Once valves 3 and 4 are opened, the N_2_-carrying CBN particles form a gas-solid two-phase flow and enter into the first-stage atomizer. The matrix alloy was heated to 1650 °C in the intermediate-frequency induction furnace. Negative pressure can be generated by the tight coupling assembly between the guide tube and first-stage atomizer. Therefore, the molten alloy flew into the atomization chamber through guide tube under the action of gravity and negative pressure. The N_2_-CBN two-phase flow was first ejected into the molten alloy. The surface tension of molten alloy was destroyed and atomized into large droplets. In the second-stage atomization, large droplets were disintegrated again by the high-speed airflow into smaller ones. CBN particles were entrapped by the liquid phase molten droplet due to the stickiness of liquid phase. The liquid phase was rapidly cooled down and solidified during the fast flight. A stiff junction between CBN particles and metal droplets was created on the contact area.

Figure 2 shows the morphology of the atomized CBN MAPs. As shown in Figure 2a, the shape of atomized MAPs is all nearly spherical. This means that the needed solidification time of atomized powders is still longer than the spheroidisation time. The MAPs have enough time to shrink into spheres within the solidification time. The spherical structure is conducive to the uniform and intensive arrays of MAPs during the polishing process. In addition, it is observed that fine CBN particles are densely embedded on the surface of MAPs (Figure 2b,c). Enough abrasives can provide an excellent polishing performance of prepared MAPs.

In addition, the morphology of CBN particles before and after was analyzed. Based on the analysis results of Figure 3a,b, the original morphology of CBN particle is maintained. The sharp cutting edge of CBN particle is not destroyed after the atomization process. This is because the predetermined temperature of the alloy matrix is lower than the melting point of CBN (3000 °C). Additionally, the rapid cooling rate of atomization process makes the liquid powder solidify in a short time. Therefore, the high temperature around CBN particles has little effect on the damage to the cutting edges. The CBN particle maintains its original finishing capability.

The XRD pattern of prepared MAPs is shown in Figure 3c. The phase of MAPs is shown as a combination of α-Fe and CBN. The rapid cooling rate of atomization process gives matrix phases very little time to nucleate and grow before solidification. The matrix is a disordered phase (α-Fe). Moreover, according to the outcome, the crystal structure of CBN particles has not changed during the atomization process.

### 2.2. Magnetic Tool Design

In the MAF process, an electromagnet can provide a strong magnetic field in the processing zone and MAPs are easily taken down from the electromagnet. However, the complex cooling system results in a bulky size of the electromagnet, which cannot provide a high rotation speed. Therefore, in the experiments, a permanent magnet (Nd-Fe-B, Bhmax/45 MGOe, made by Xunjie company, Jinan, Shandong, China) was selected as a magnetic field generator. Then the MAPs are arranged along the magnetic line of force and formed a cutting tool. The magnetic force (F) acting on the MAPs under a magnetic field is as follows:(1)$$F\sim d^{3}X_{r}H\frac{\partial H}{\partial t}$$
where, *d* and $X_{r}$ are the size and magnetic susceptibility of MAPs, respectively. *H* and $\frac{\partial H}{\partial t}$ are the magnetic field intensity and magnetic field intensity gradient at the location of MAPs, respectively.

From Equation (3), the magnetic force of MAPs is proportionate to the magnetic field intensity of the magnetic field. In order to improve the magnetic field intensity, the permanent magnet was designed to conduct groove openings. The purpose of the grooves openings was to reduce the circumference area of the permanent magnet (H = Φ/A, where, Φ is the magnetic flux quantity, and A is the circumference area of the magnet) [21]. To optimize the number and dimension of grooves, a 3-D model containing magnetic pole, working gap and workpiece was simulated by finite element analysis-based software package Ansoft Maxwell. The geometry of the model and the parameters used here are given in Table 1.

As shown in Figure 4, the simulation results confirm that groove openings can increase the magnetic flux density. The maximum magnetic flux density (0.338 T) of cylindrical magnet with grooves is larger than that of cylindrical magnet (0.289 T). In addition, grooves change the distribution of magnetic flux density. For cylindrical magnet only two discontinuous magnetic regions have high magnetic flux density, while the groove provides six uniform areas with higher magnetic flux density. The MAPs will have high circular velocities under the distribution. Comparing the simulation results of different dimensions of rectangle grooves, the optimum design (Number 6 × Width 2 mm × Depth 2 mm) was obtained.

### 2.3. Experimental Setup

To understand the finishing characteristic of CBN MAPs in the MAF process, the experiments were carried out on a vertical milling machine. Figure 5a shows the developed magnetic field generator. The magnet pole is fixed in the spindle chuck and rotated with the spindle. The gap between the magnet and workpiece is called working gap which is filled with a certain amount of atomized CBN MAPs. These MAPs can form a flexible magnetic abrasive brush (MAB) with certain stiffness on the surface of the workpiece under a magnetic field. The experimental setup is shown in Figure 5b.

At the beginning of the MAF process, the normal force of MAP can cause CBN particles to create an indentation depth on the wearing surface, as shown in Figure 6. The normal pressure ($P_{n}$) of CBN particle is calculated by Equation (2) as follows [22]:(2)$$P_{n}=\frac{B^{2}}{4\mu_{0}}\cdot \frac{3\pi \left(\mu_{iron}-1\right)\delta }{3\left(2+\mu_{iron}\right)+\pi \left(\mu_{iron}-1\right)\delta }$$
where, B is the magnetic flux density, $\mu_{0}$ is the vacuum permeability,$\;\mu_{iron}$ is the magnetic permeability of the iron matrix, δ is the volume fraction of the iron matrix.

Based on Hertz’s contact theory, the indentation depth (h) can be calculated by Equation (3) as follows:(3)$$h=\frac{1}{2}{\left(\frac{9}{16}\cdot \frac{1}{\frac{d_{g}}{2}}\right)}^{\frac{1}{3}}{\left(\frac{p_{n}}{E_{c}}\right)}^{\frac{2}{3}}$$
where, $d_{g}$ is the diameter of CBN particle. $E_{c}$ is the elastic modulus between particle and workpiece.

It shows that a larger magnetic flux density will produce a greater h on the surface. In the MAF process, the CBN particle will experience normal force, tangential cutting force and centripetal force. As the MAPs rotated with the magnetic pole, the tangential cutting force will remove the material of indentation. The amount of material removed is very small. Therefore, the MAF process can provide a remarkable improvement in surface roughness without changing the profile of parts.

### 2.4. Workpiece Preparation

The workpiece material is a kind of cemented carbide with high hardness. The value of hardness was measured by a Microhardness Tester (FM-800, made by Hengjian company, Jinan, Shandong, China). The chemical composition of the work material was obtained from the Energy Dispersive Spectrometer (EDS). The detail of work material is shown in Table 2.

## 3. Experiment Details

In the MAF process, the properties of MAP, the mechanical properties of the work material and different process parameters affect the surface quality of the workpiece. For relatively soft materials, the workpiece can be easily finished with a magnetic abrasive tool. However, hard materials should require a stronger abrasive tool to remove the surface material. As shown in the fishbone diagram (Figure 7), the surface finish is mainly affected by the following process parameters.

The design experiment is very important for this study. In order to obtain a response relationship between process parameters and surface finish, a central composite design (CCD) experiment was applied. The CCD can fit a second-order function relation between variable factors and response value. According to Figure 6, the working gap, the rotational speed of the magnetic pole, the feed rate of the workpiece and the mesh number of MAPs were selected as variable factors. Pilot experiments for each factor were performed to determine a suitable range of process parameters.

The range of the working gap was selected based on the magnetic field intensity. The surface roughness of cemented carbide polished under a greater 3 mm working gap had a little change. The upper limit of the working gap was selected as 3 mm. The rotational speed of the magnetic pole will affect the stability of the formed magnetic abrasive tool. Small rotational speed limits the finishing efficiency of the MAF process, while high rotational speed makes the MAPs leave the working gap. Thus, the range of rotational speed was 1000–2000 rpm. A low feed rate would mean a high processing time for the workpiece, while a high feed rate would decrease the finishing effectiveness. For polishing hard materials, the upper limit of feed rate was selected at a relatively small value (5 mm/min). The mesh number was selected from coarse to fine size of atomized MAPs. Table 3 shows the five levels of the selected four factors in the experiment. The quantity of MAPs can affect the stability and stiffness of the abrasive tool at a certain working gap. According to the value of the working gap and the surface area of the magnetic pole, the optimal weight of MAPs used at different working gaps was determined, as shown in Table 4. To reduce the friction heat, SAE 5W-30 oil lubricant provided by Shandong Hanneng Company was applied in the MAF process. In total, 0.2 g of oil lubricant was used in each orthogonal experiment.

Since the initial surface roughness is different at the beginning of each experiment, it is incorrect to select the finished roughness as the response value. To compensate for this in each experiment, the initial surface roughness of samples was ground and controlled between 0.3 µm and 0.4 µm. In addition, we measured the surface roughness of three areas and used the average value as the initial surface roughness. The roughness value of the same three areas after polishing was averaged as the final surface roughness. Therefore, a variable of the percentage change in $R_{a}$ ($\%\Delta R_{a})$, was set as the response output in this experiment, and this was calculated by Equation (4). The detail of MAF experimental results is shown in Table 5.
(4)$$\%\Delta R_{a}=\frac{\mathrm{initial}\;\mathrm{surface}\;\mathrm{roughness}-\mathrm{final}\;\mathrm{surface}\;\mathrm{roughness}}{\mathrm{initial}\;\mathrm{surface}\;\mathrm{roughness}}\times 100$$

## 4. Results and Discussions

### 4.1. Statistical Model of %ΔR_a_

Table 6 shows the details of the arrangement of the parameter combinations, measured values of surface roughness and $\%\Delta R_{a}$ calculated by Equation (3). Using analysis of variance, a regression equation for predicting $\%\Delta R_{a}$ within the given region was obtained. The developed equation was complicated due to the linear effect, quadratic effect and interaction effect of the selected factors on the response value. In order to simplify and improve the regression equation, many insignificant terms have been eliminated. The analysis result is presented in Table 6, and a new equation containing significant terms is shown in Equation (5).
(5)$$\%\Delta R_{a}=-108.231-6.81417X_{1}+0.175905X_{2}+20.64167X_{3}+\;0.456633X_{4}-4.99667X_{1}^{2}-5.21E05X_{2}^{2}-2.23917X_{3}^{2}-0.00142X_{4}^{2}+\;0.0078X_{1}X_{2}+3.845X_{1}X_{3}-0.03115X_{1}X_{4}-0.01273X_{2}X_{3}$$

### 4.2. Interaction Effects of Process Parameters

Figure 8 shows the mean response result of four parameters on the change of surface roughness ($\%\Delta R_{a}$). The $\%\Delta R_{a}$ decreases with increasing working gap. A certain value of rotational speed (1500 rpm) produces a high $\%\Delta R_{a}$ on the surface. As the feed rate increases, the $\%\Delta R_{a}$ is getting worse and worse. A certain mesh number of MAPs results in a high $\%\Delta R_{a}$ during the MAF process. According to the analysis of variance, some significant interaction effects of two parameters affect the value of $\%\Delta R_{a}$. The analysis data reveal different trends. Three significant groups of interaction effect were plotted in 3D curved surfaces. To provide a clearer interaction relationship, the 2D graph was also plotted by using Equation (5).

As seen in Figure 9, under different rotational speeds, a lower working gap causes a higher $\%\Delta R_{a}$. Based on the simulation results of the cylindrical magnetic pole, the magnetic flux density on the workpiece surface was decreased with an increased working gap. For higher working gaps, the magnetic lines of forces become loose and the strength of the formed abrasive tool is weak. The available magnetic force of MAPs decreases, causing CBN particles insufficient indentation depth on cemented carbide. The original burrs and scratches still remain on the workpiece surface, which results in a low $\%\Delta R_{a}.$ Conversely, at a low working gap, the large magnetic flux density can provide a high indentation of CBN particles to remove sufficient material from the surface. Moreover, the value of $\%\Delta R_{a}$ increases first with the increasing rotational speed, the decrease appears after a certain rotational speed. The rotational speed determines the collision number between CBN particles and the material surface, coupled with the stability of the abrasive tool. The small $\%\Delta R_{a}$ produced at a relatively slow rotational speed is caused by an insufficient collision.

In addition, the low $\%\Delta R_{a}$ is obtained at a relatively high rotational speed. Figure 10 shows the distribution of bonded MAPs in a magnetic field. The bonded MAPs will be magnetized and aligned along the magnetic line. During the process, the holding force of the magnetic force line on MAPs is sufficient under a slow rotational speed due to the strong magnetic flux density. The sufficient centripetal force of the MAP could prevent the formed abrasive chain from being disarranged. Conversely, the little holding capability of the magnetic chain could not meet enough centripetal force of MAPs at a high rotational speed, which appears as a decrease in $\%\Delta R_{a}$.

Figure 11 shows that, under different mesh numbers, $\%\Delta R_{a}$ decreases as the working gap increases. The maximum $\%\Delta R_{a}$ is found at a certain mesh number (150) with a 1.0 working gap. At different working gaps, coarse MAPs can maintain a more stable cutting force than fine MAPs due to the high amount of iron matrix. Therefore, CBN particles can easily remove the material from the surface. However, the use of coarse MAPs reduces the effective contact area between MAPs and the workpiece. The reduction of active hard particles cannot guarantee that the whole surface will be polished, so the surface finish is not good. Fine MAPs increase the number of abrasive chains and the available contact area of the magnetic abrasive tool. This will be helpful for adding more active hard particles in the grinding process. However, a small amount of iron matrix may affect the tangential cutting force of fine MAPs. The lack of cutting force makes it impossible to remove the indentation material completely. Thus, the low grinding energy and insufficient cutting force of fine MAPs result in low $\%\Delta R_{a}$. Furthermore, the results revealed that the surface roughness produced by MAPs with coarse sizes of between 50 and 100 mesh is better than that of fine sizes (between 200 and 250 mesh) under different working gaps. This shows that, compared with an effective contact area, a sufficient tangential cutting force of MAPs can more effectively remove material from the surface.

As shown in Figure 12, under different rotational speeds, a low feed rate can provide a high $\%\Delta R_{a}$. During the MAF process, the movement distance of the workpiece relative to hard particles in unit time is determined by the feed rate. The low feed rate ensures that CBN particles have enough time to remove material from the polishing trajectory. Moreover, the low feed rate can provide sufficient collision times between MAPs and burrs present on the surface. Therefore, under different rotational speeds, a low feed rate can produce a better surface roughness. The rotational speed of the magnetic pole is synchronized with the instantaneous grinding speed of hard particles due to the holding force of the magnetic induction line. From Figure 12, at a low feed rate, the high rotational speed produced a low $\%\Delta R_{a}$. This may be because the insufficient centripetal force of MAPs causes the dispersion of abrasive chains. A sufficient indentation depth cannot be produced on the finished surface, which results in a poor surface finish. At low rotational speeds, the $\%\Delta R_{a}$ was also observed at relatively small values. This may be because the low rotational speed cannot provide enough collision numbers between CBN particles and workpieces to remove material from the surface.

### 4.3. Validation Test

In this study, five groups of process parameters were selected for verification experiments of the developed regression equation, the analysis result is shown in Table 7. Further, the optimal parameter combination was figured out through the solution of the quadratic equation in Design-Expert software. The value of the combination was a 1 mm working gap, 1638 rpm, 1 mm/min feed rate and 150 mesh number. In order to consider the precision of the regression equation on the optimal parameter combination, three repeated verification tests were performed on the workpiece. The theoretical prediction was calculated using Equation (5), and the actual $\%\Delta R_{a}$ was measured by the MicroXAM-100 white light interferometer (Z-resolution 16 nm, KLA Tencor, Houston, USA). Figure 13 shows the variation in the surface profiles.

### 4.4. Surface Quality of Cemented Carbide

For better investigating the finishing performance of atomized CBN MAPs, the surface morphology of the workpiece was performed by scanning electron microscopy (SEM, Kexin company, Beijing, China). As shown in Figure 14a, many burrs and grinding marks exist on the original surface. After the MAF process employing the working gap of 1 mm, 2500 rpm, feed rate of 1 mm/min and 150 mesh, it can be observed that some original grinding marks still exist, as seen in Figure 14b. Too high a rotational speed affects the stability of the abrasive chain and results in the ineffective finishing of MAPs. Therefore, the original grinding marks could not be completely removed from the workpiece surface. After the MAF process employing the working gap of 2 mm, 1638 rpm, feed rate of 1 mm/min and 150 mesh, the finished surface still presents a high roughness, as seen in Figure 14c. This is because, at high working gaps, CBN MAPs could not produce sufficient indentation depth on the surface, which results in a low material removal. Comparing the SEM images of Figure 14b,c, Figure 14d shows that atomized CBN MAPs have effectively removed scratches and burrs from the surface. Under a low working gap, CBN particles produced a uniform and optimal indentation depth on the surface. The sufficient cutting force of MAPs removed material from the surface. No excessive scratches were found. Thus, the atomized CBN MAPs can give the hard material cemented carbide a nano-level surface finish.

### 4.5. Comparison of Polishing Performance of MAPs

Table 8 shows the polishing performance of the MAPs prepared by different technologies. It can be seen that the MAF process can produce excellent surface quality on different materials. However, the present work prepared CBN MAPs can achieve a nanometer roughness on cemented carbide. The hardness of the workpiece seems to be the highest among the MAPs referenced. The high wear resistance and hardness of CBN enhance the finishing capacity of MAPs.

## 5. Conclusions

In this study, to better improve the finishing performance of MAP, the super-hard cubic boron nitride (CBN) tiny powder was selected as an abrasive and CBN MAPs were successfully fabricated using an atomization technology. The atomized MAPs have a spherical morphology and CBN particles are densely embedded on the surface of the MAP. A hard material, cemented carbide (864 Hv), was selected as the test workpiece.

Based on the analysis results of the central composite design (CCD) experiment, at low working gap MAPs produced a good surface finish on the workpiece because high magnetic flux density could provide an optimum indentation depth of CBN particles on the surface. The abrasive chains maintained excellent stability at a high rotational speed (1500 rpm) of the magnetic pole. The sufficient tangential cutting force of CBN particles effectively removed the original material from the surface. The regression model for the change of surface roughness was developed. Through the result of the validation test, the errors between prediction and measurement are below 10%.

The high bonding strength and cutting edge of CBN particles produced a remarkable improvement in the surface roughness (57.6 nm) of cemented carbide. Atomized CBN MAPs display excellent finishing performance on hard materials.

## Figures and Tables

**Figure 1 micromachines-13-02167-f001:**
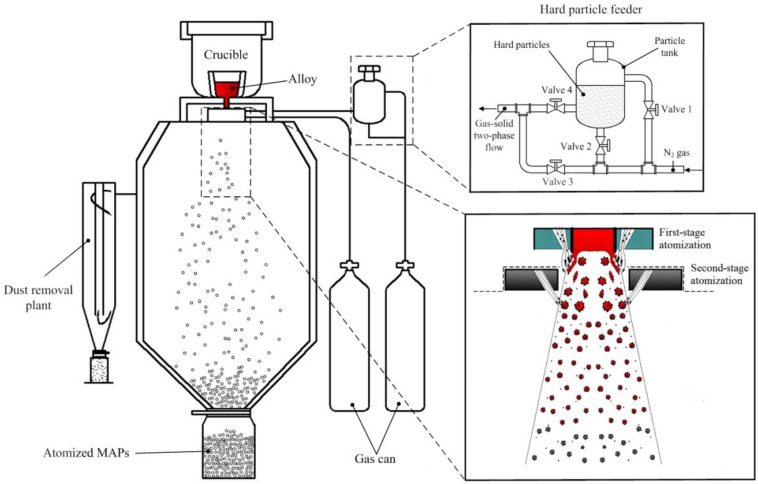
Schematic diagram of MAPs prepared by double-stage atomization.

**Figure 2 micromachines-13-02167-f002:**
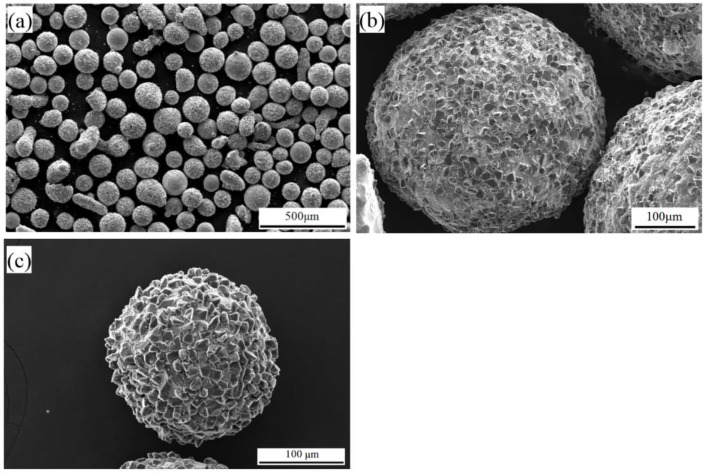
(**a**) SEM micrographs of the prepared MAP; (**b**) size with mesh no. 50; (**c**) and size with mesh no. 100.

**Figure 3 micromachines-13-02167-f003:**
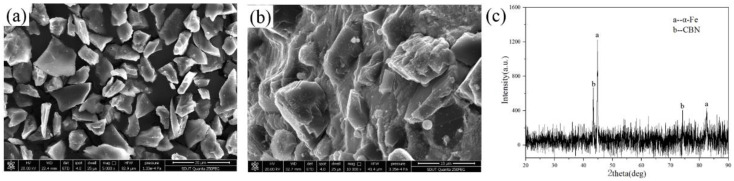
SEM micrographs of CBN particles (**a**) before atomization; (**b**) after atomization; (**c**) XRD and phase identification for MAPs.

**Figure 4 micromachines-13-02167-f004:**
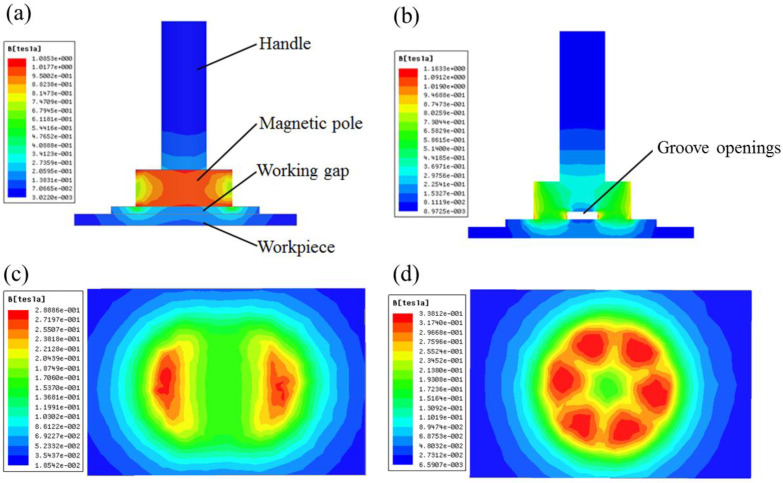
Simulation results: (**a**) model of cylindrical magnetic pole; (**b**) model of cylindrical magnetic pole with grooves; (**c**) magnetic flux density on workpiece surface performed at model (**a**); (**d**) magnetic flux density on workpiece surface performed at model (**b**).

**Figure 5 micromachines-13-02167-f005:**
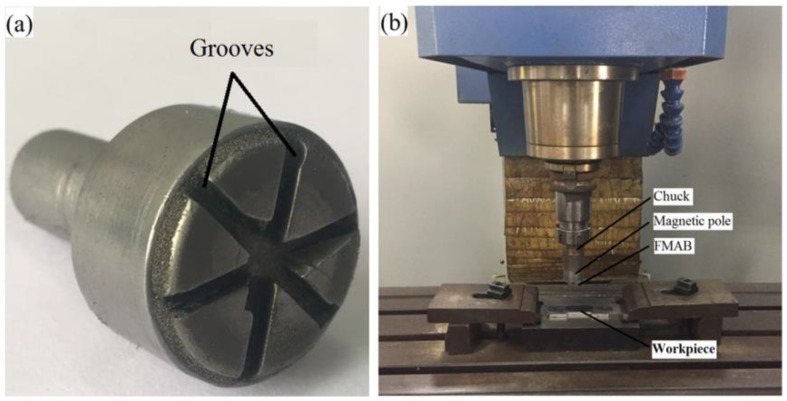
(**a**) Magnetic pole; (**b**) magnetic abrasive finishing setup.

**Figure 6 micromachines-13-02167-f006:**
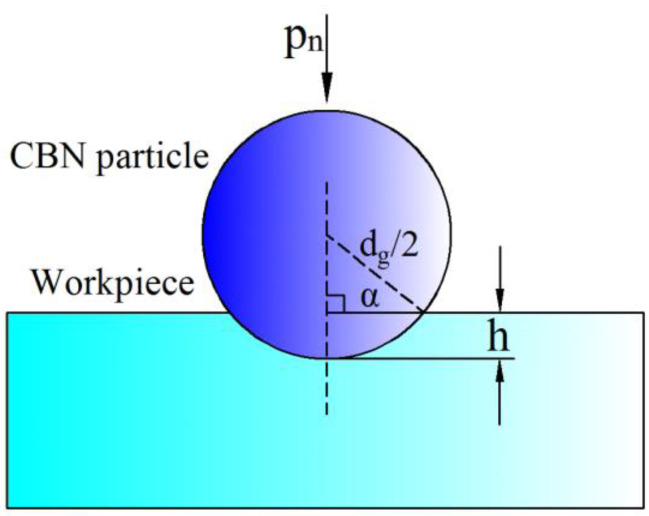
Schematic of indentation depth for a CBN particle.

**Figure 7 micromachines-13-02167-f007:**
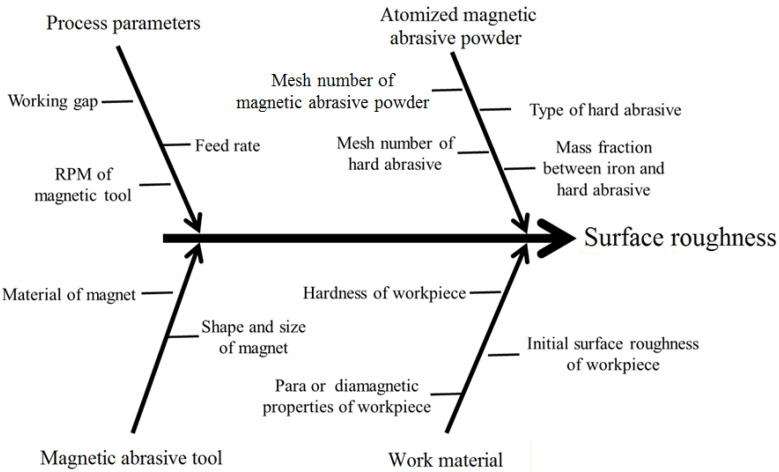
Fishbone diagram showing important process parameters.

**Figure 8 micromachines-13-02167-f008:**
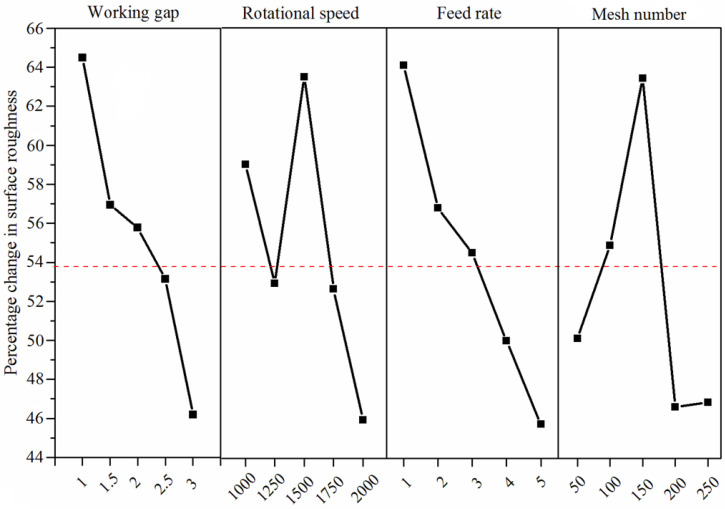
Mean response of process parameter on $\%\Delta R_{a}$.

**Figure 9 micromachines-13-02167-f009:**
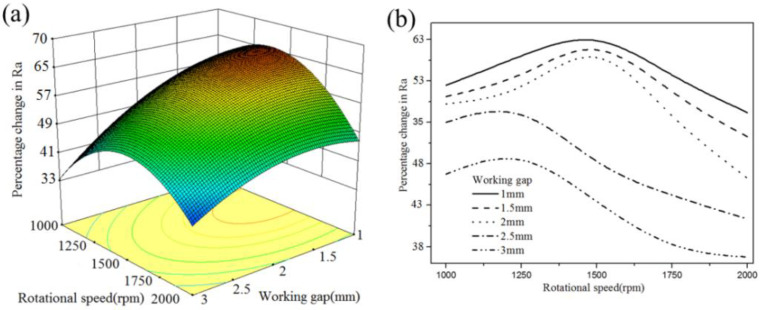
Interaction effect between the working gap and rotational speed (feed rate: 1 mm/min; mesh no.150): (**a**) 3D curved surface; (**b**) 2D graph.

**Figure 10 micromachines-13-02167-f010:**
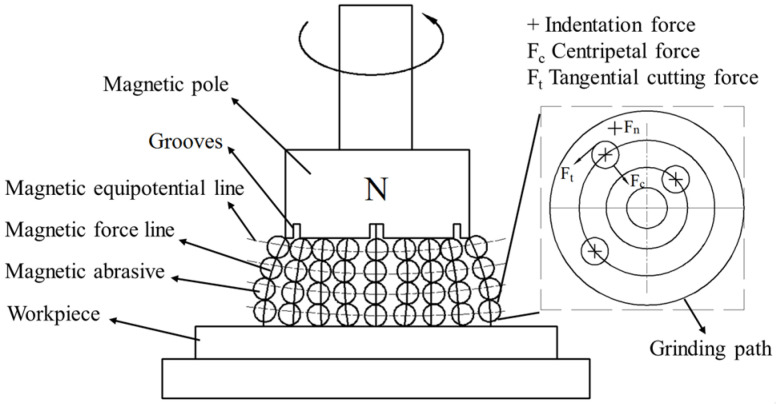
MAF schematic of polishing planar workpiece.

**Figure 11 micromachines-13-02167-f011:**
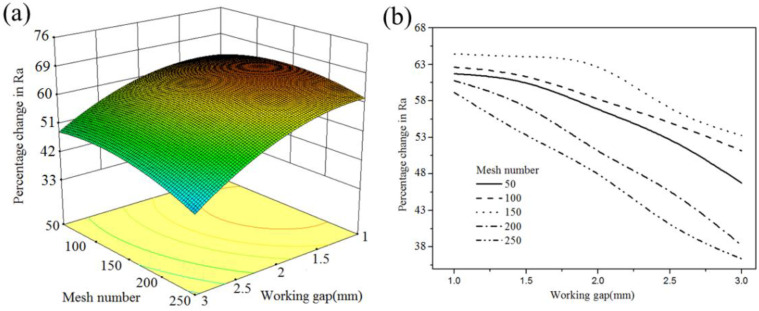
Interaction effect between the working gap and mesh number (feed rate: 1 mm/min; rotational speed: 1500): (**a**) 3D curved surface; (**b**) 2D graph.

**Figure 12 micromachines-13-02167-f012:**
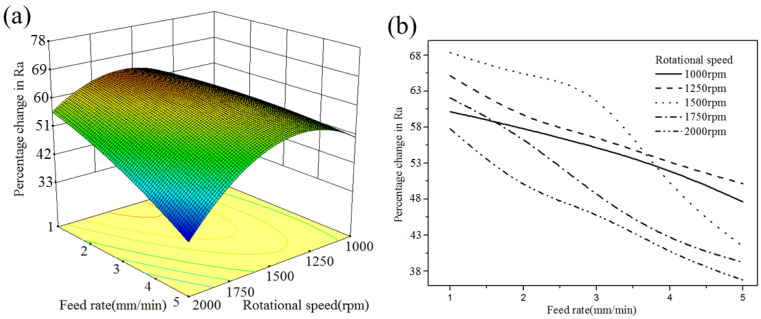
Interaction effect between the feed rate and rotational speed (working gap: 1 mm; mesh no.150): (**a**) 3D curved surface; (**b**) 2D graph.

**Figure 13 micromachines-13-02167-f013:**
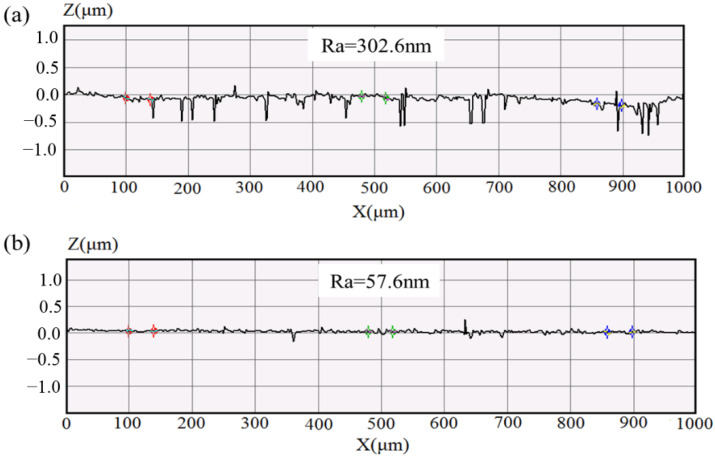
Surface roughness for (**a**) original surface; (**b**) finished surface when using 1 mm working gap, 1638 rpm, 1 mm/min feed rate and 150 mesh.

**Figure 14 micromachines-13-02167-f014:**
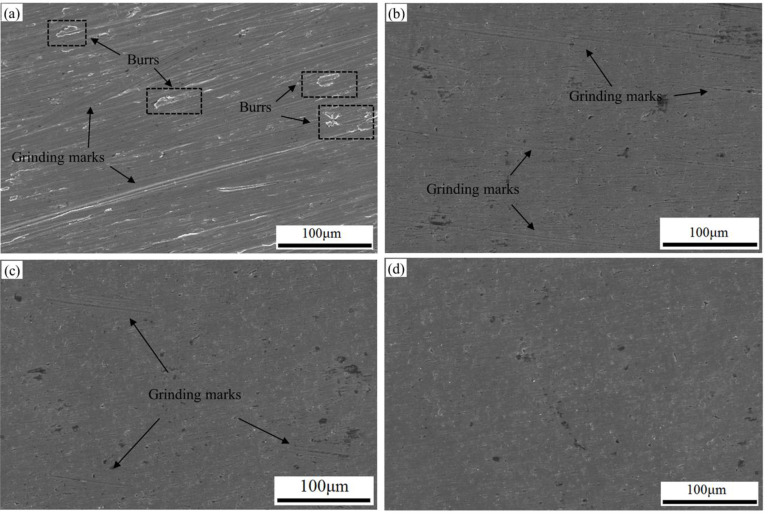
SEM micrographs of (**a**) original surface; (**b**) finished surface when employing 1 mm working gap, 2500 rpm, 1 mm/min and 150 mesh in MAF; (**c**) finished surface when employing 2 mm working gap, 1638 rpm, 1 mm/min and 150 mesh in MAF; (**d**) finished surface when employing 1 mm working gap, 1638 rpm, 1 mm/min and 150 mesh in MAF.

**Table 1 micromachines-13-02167-t001:** Material and geometry of the model.

Model	Dimension	Material	Relative Permeability
Handle	Φ15 mm × H30 mm	Stainless steel	1
Working gap	50 mm × 30 mm × 2 mm	Air	1.0000004
Workpiece	70 mm × 30 mm × 5 mm	Cemented carbide	M-H curve
Magnetic pole	Φ30 mm × H20 mm	N45	1.2
Groove openings	Width: 1–3 mm Depth: 1–3 mm Number: 4–8		

**Table 2 micromachines-13-02167-t002:** The details of test workpiece.

Hardness	864 Hv
Dimension	15 mm × 15 mm × 5 mm
Alloying element	Wt.%
W	64.45
Ti	14.55
C	9.16
O	5.32
Co Cr	3.88 2.64

**Table 3 micromachines-13-02167-t003:** Parameters and levels for experiment.

Factors	Parameter	Level				
		−2	−1	0	1	2
X_1_	Working gap (mm)	1	1.5	2	2.5	3
X_2_	Rotational speed (rpm)	1000	1250	1500	1750	2000
X_3_	Feed rate (mm/min)	1	2	3	4	5
X_4_	Mesh number	50	100	150	200	250

**Table 4 micromachines-13-02167-t004:** Weight of MAP applied in different working gaps.

Working Gap (mm)	Quality of MAP (g)
1	2
1.5	2.5
2	3
2.5	3.5
3	4

**Table 5 micromachines-13-02167-t005:** Details of experimental results of %Δ*R_a_* for workpiece.

Run Order	Working Gap (mm)	RPM	Feed Rate (mm/min)	Mesh Number	Finished Roughness (nm)	$\%\Delta R_{a}$
1	2	1500	1	150	101.6	68.31
2	2	1500	3	150	122.2	63.86
3	2.5	1250	4	200	207.2	46.61
4	2.5	1750	2	100	165.7	54.96
5	1.5	1250	4	200	175.2	53.54
6	2	1000	3	150	155.8	55.23
7	2.5	1750	4	200	233.6	36.7
8	1.5	1250	2	100	118.5	63.54
9	1.5	1250	4	100	183	52.96
10	2.5	1750	4	100	198.6	45.15
11	2	2000	3	150	176.8	46.27
12	2.5	1250	2	200	237.8	37.26
13	2	1500	5	150	197.3	41.44
14	2	1500	3	250	198.5	48.57
15	1.5	1750	2	200	138.7	61.7
16	1.5	1250	2	200	168.7	57.39
17	2.5	1250	4	100	183	50.8
18	2.5	1750	2	200	173.1	51.24
19	1.5	1750	4	100	197.7	52.01
20	1	1500	3	150	119.5	64.43
21	2	1500	3	50	189.7	50.73
22	2.5	1250	3	150	209.4	46.57
23	3	1500	3	150	196.3	43.26
24	1.5	1750	2	100	150	61.15
25	1.5	1750	4	200	184.4	43.6

**Table 6 micromachines-13-02167-t006:** Analysis of variance (ANOVA) for the obtained Equation (4) without insignificant terms.

Source	DF	Seq.SS	MS	F	p	R^2^	
Regression	12	1885.805	134.7	20.378	0.000	95.005%	$F_{\left(0.05,12,15\right)}^{standard}=2.48$ $F^{standard}>F_{\left(0.05,10,15\right)}^{standard}$ $F_{\left(0.05,10,15\right)}^{standard}=2.54$ $F^{lack-of-fit}<F_{\left(0.05,10,15\right)}^{standard}$ Model is adequate and lack of fit is insignificant
Linear	4	745.57					
Square	4	1181.078					
Interaction	4	217.623					
Residual error	15	99.15	6.61				
Lack-of-fit	10	89.665		2.31	0.0502		
Pure error	5	9.486					
Total	27	1984.957					

**Table 7 micromachines-13-02167-t007:** Validation details of statistical model.

Order	Process Parameters				*R_a_*		$\%\Delta R_{a}$		
	$X_{1}$	$X_{2}$	$X_{3}$	$X_{4}$	Original *R_a_* (nm)	Finished *R_a_* (nm)	Predicted Values	Experimental Observation	% Error
1	1	2000	1	200	348.2	140.3	64.05	59.7	7.28
2	1.5	1000	5	50	316.5	130	55.33	58.93	6.1
3	2	1750	4	100	365.7	205.8	47.33	43.72	8.25
4	2.5	1500	3	150	374	173.2	57.57	53.69	7.23
5	3	1250	2	250	355.3	156.5	59.95	55.95	7.15
R1	1	1638	1	150	346.1	71	74.35	79.49	6.47
R2	1	1638	1	150	325	93.8	74.35	71.14	4.5
R3	1	1638	1	150	302.6	57.6	74.35	80.96	8.15

**Table 8 micromachines-13-02167-t008:** Performance survey of MAPs.

Preparation Technology	MAPs	Workpiece Material	Original *R_a_*	Finished *R_a_*	$\%\Delta R_{a}$
Mixing [3]	Fe/Al_2_O_3_	Ti-6Al-4V	1.121 μm	0.046 μm	95.9%
Sintering [23]	Fe/Al_2_O_3_	Steel718	2.8 μm	0.39 μm	86%
Gel [16]	Fe/SiC	SKD11	0.65 μm	0.110 μm	83%
Plasma sprayed [24]	Fe/Al_2_O_3_	SS316	0.299 μm	0.068 μm	77%
Alloy-Hardening [25]	CI-MAPs	Zr-alloy	0.361 μm	0.085 μm	76.5%
Present work	Fe/CBN	cemented carbide	0.3 μm	0.0576 μm	80.8%

## Data Availability

This study did not report any data.

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
