# Peer review of "Polishing Characteristics of Cemented Carbide Using Cubic Boron Nitride Magnetic Abrasive Powders"

_micromachines, 2022, doi:10.3390/mi13122167_

Round 1
Reviewer 1 Report
1. Micromachines is paying more and more attention to the quality of articles, especially originality and innovation. The authors should state the innovation of the article more clearly and fully in the Abstract, Introduction, Results and discussion and Conclusion sections.
2. The novelty/achievements of this work should be highlighted.
3. Please provide more details on the experimental analysis of the solids and liquids along with experimental errors etc.
4. The authors should make a comparison with the results in recent publications.
5. In Results and discussion section, the analysis is unclear and do not cite related references to further in-depth present the innovativeness and significance in this work. Please revise this section.
6. How about the microstructure performance in cubic boron nitride? I suggest the authors to add XRD test.
7. Finally, the English of the manuscript need to improve. Please ask a native English speaker to revise and proofread their revised manuscript before re-submission.
Author Response
Ref.No.:Micromachines -2066800
Title: Polishing characteristics of cemented carbide using cubic boron nitride magnetic abrasive powders
Journal: Micromachines
We would like to express our sincere thanks to the reviewer for the constructive and positive comments.
Detailed response to Reviewer(s)' Comments:
- Micromachines is paying more and more attention to the quality of articles, especially originality and innovation. The authors should state the innovation of the article more clearly and fully in the Abstract, Introduction, Results and discussion and Conclusion sections.
Response:
Thanks a lot for your valuable review. We are very sorry for ignoring the emphasis on the innovation of our original manuscript. The innovation of this study is to demonstrate the excellent polishing performance of MAPs prepared by atomization on superhard materials. Although the prepared Al2O3 MAPs have outstanding finishing and service life characteristics, the weak wear resistance and hardness of abrasive phase make produced MAPs perform poorly in terms of material removal efficiency on high hardness workpiece (ceramics, cemented carbide, etc). The original grinding marks and fault layers are still visible on the polished surface. Therefore, in this study, CBN micro particles with high wear resistance and hardness were used as abrasive phase to enhance the polishing performance of MAPs. According to the findings of this research, the atomized CBN MAPs can be an effective polishing tool for finishing the hard materials.
In our revised manuscript, the innovation of this study was stated. For example, the revised abstract is:
“…In order to improve the poor finishing performance and short service life of MAPs in polishing super-hard materials, a double-stage atomization technique was used to successfully manufacture MAPs with a CBN as abrasive phase. The prepared results show that CBN abrasives with their original structure were deeply and densely embedded on the surface of spherical MAPs. … According to MAF results, the strong cutting ability of atomized CBN MAPs improved the surface roughness of cemented carbide over 80% at the optimum parameters. The strong cutting ability of atomized CBN MAPs can produce good surface quality on the hard materials. The findings of this research can promote a large-scale application of MAF technology in the surface polishing of hard materials.”
The revised introduction is:
“The weak wear resistance and hardness of abrasive make developed MAPs perform poorly in the material removal efficiency. In order to improve the finishing performance of MAP, super-hard cubic boron nitride (CBN) fine particle was selected as abrasive. CBN abrasives and metal liquid were atomized by a double-stage atomization. MAF test was done to evaluate the polishing performance of atomized MAPs on cemented carbide.”
Other partial modifications have been marked in red in our revised manuscript.
- The novelty/achievements of this work should be highlighted.
Response:
The achievements of this word is to provide an effective polishing tool for polishing the hard materials. An atomization technology was used to prepare CBN MAPs. According to MAF experiments, the atomized CBN MAPs produce good surface quality on the cemented carbide. In our revised manuscript, the abstract, introduce and other parts of the paper were revised and marked in red.
- Please provide more details on the experimental analysis of the solids and liquids along with experimental errors etc.
Response:
Thank you for your valuable comments. The analysis of experiment result was added in the revised manuscript. For example, the more details in the working gap and rational speed was revised as follow:
“As seen in Fig.9, under different rotational speeds, a lower working gap causes a higher. Based on simulation results of cylindrical magnetic pole, the magnetic flux density on the workpiece surface was decreased with an increase working gap. For higher working gap, the magnetic lines of forces become loose and the strength of formed abrasive tool is weak. The available magnetic force of MAPs decreases causing CBN particles insufficient indentation depth on cemented carbide. The original burrs and scratches still remain on the workpiece surface which results a low Conversely, at a low working gap, the large magnetic flux density can provide a high indentation of CBN particles to remove sufficient material from the surface. Moreover, the value of increases first with the increasing rotational speed, the decrease appears after a certain rotational speed. The rotational speed determines the collision number between CBN particles and material surface coupled with the stability of abrasive tool. The small produced at relatively slow rotational speed is caused by the insufficient collision.”
- The authors should make a comparison with the results in recent publications
Response:
Table 8 is a brief of literature survey on MAF results that were focused on different materials by the MAPs prepared by different processes. The detailed description of comparison with the polished results is presented in section "5.5 Comparison of polishing performance of MAPs" and Table 8.
- In Results and discussion section, the analysis is unclear and do not cite related references to further in-depth present the innovativeness and significance in this work. Please revise this section.
Response:
Thanks for your advice, and I have rewritten the section “Results and discussions”. The modifications made in the revised manuscript have been highlighted by RED text.
- How about the microstructure performance in cubic boron nitride? I suggest the authors to add XRD test.
Response:
Thanks for your valuable advice. A new Fig.3 was added in the paper. The supplementary content is as follows:
“This means that the needed solidification time of atomized powders is still longer than the spheroidisation time. The MAPs have enough time to shrink into sphere within the solidification time. The spherical structure is conductive to the uniform and intensive arrays of MAPs during the polishing process. In addition, it is observed that fine CBN particles are densely embedded on the surface of MAPs (Fig. 2 (b) and (c)). Enough abrasives can provide an excellent polishing performance of prepared MAPs.
In addition, the morphology of CBN particle before and after was analyzed. Based on the analysis results of Fig. 3 (a) and (b), the original morphology of CBN particle is maintained. The sharp cutting edge of CBN particle are not destroyed after the atomization process. This is because the predetermined temperature of the alloy matrix is lower than the melting point of CBN (3000℃). Additionally, the rapid cooling rate of atomization process makes the liquid powder solidify in a short time. Therefore, the high temperature around CBN particles have little effect on the damage to the cutting edges. The CBN particle maintain its original finishing capability.”
Figure 3. SEM micrographs of CBN particles (a) before atomization; (b) after atomization; (c) XRD and phase identification for MAPs.
- Finally, the English of the manuscript need to improve. Please ask a native English speaker to revise and proofread their revised manuscript before re-submission.
Response:
We have revised the whole manuscript carefully and tried to avoid any grammar or syntax error. In addition, we have asked several colleagues who are skilled authors of English language papers to check the written English.

Reviewer 2 Report
1. The title of the paper is suggested to be changed, because most of the content of the paper describes the magnetic abrasive finishing experiment, and there are few rules about the essential effects of BCN.
2. It is suggested to revise and improve the abstract of the paper. From the abstract, we can only see what has been done in the paper, and there is a lack of highlights.
3. Is there any literature report on CBN magnetic grinding powder at home and abroad? We cannot see this information in the introduction.
4. It is recommended to supplement the original BCB powder topography photos.
5. The distribution uniformity of CBC particles certainly affects the experimental results. Please supplement the distribution photos of CBN particles in MAP powder.
6. In "5.2. Interaction effects of process parameters", the analysis and interpretation of experimental results in Figure 7 and Figure 8 are lacking.

Author Response
Ref.No.:Micromachines -2066800
Title: Polishing characteristics of cemented carbide using cubic boron nitride magnetic abrasive powders
Journal: Micromachines
We would like to express our sincere thanks to the reviewer for the constructive and positive comments.
Detailed response to Reviewer(s)' Comments:
- The title of the paper is suggested to be changed, because most of the content of the paper describes the magnetic abrasive finishing experiment, and there are few rules about the essential effects of BCN.
Response:
Thanks for your valuable advice. The title has been modified to “Polishing characteristics of cemented carbide using cubic boron nitride magnetic abrasive powders”.
- It is suggested to revise and improve the abstract of the paper. From the abstract, we can only see what has been done in the paper, and there is a lack of highlights.
Response:
In our revised manuscript, the highlights of this study was stated. The revised abstract is:
“…In order to improve the poor finishing performance and short service life of MAPs in polishing super-hard materials, a double-stage atomization technique was used to successfully manufacture MAPs with a CBN as abrasive phase. The prepared results show that CBN abrasives with their original structure were deeply and densely embedded on the surface of spherical MAPs. … According to MAF results, the strong cutting ability of atomized CBN MAPs improved the surface roughness of cemented carbide over 80% at the optimum parameters. The strong cutting ability of atomized CBN MAPs can produce good surface quality on the hard materials. The findings of this research can promote a large-scale application of MAF technology in the surface polishing of hard materials.”
- Is there any literature report on CBN magnetic grinding powder at home and abroad? We cannot see this information in the introduction.
Response:
Through survey the literature at home and abroad, the report of preparation of CBN MAPs is very few. The recently paper about CBN MAPs was published by our research group. The research is about the study of CBN MAPs on polishing titanium alloy. The source of the article is “https://doi.org/10.1007/s00170-020-05810-z”.
4. It is recommended to supplement the original BCB powder topography photos.
Response:
The SEM micrograph of original CBN particles was added in the revised manuscript, as shown in Fig.3(a).
Figure 3. SEM micrographs of CBN particles (a) before atomization; (b) after atomization; (c) XRD and phase identification for MAPs.
- The distribution uniformity of CBC particles certainly affects the experimental results. Please supplement the distribution photos of CBN particles in MAP powder.
Response:
Thanks for your valuable advice. A new Fig.3 was added in the paper. Combined with Fig.2 and Fig.3, we analyzed the distribution of CBN particles on the surface of MAPs. The supplementary content is as follows:
“This means that the needed solidification time of atomized powders is still longer than the spheroidisation time. The MAPs have enough time to shrink into sphere within the solidification time. The spherical structure is conductive to the uniform and intensive arrays of MAPs during the polishing process. In addition, it is observed that fine CBN particles are densely embedded on the surface of MAPs (Fig. 2 (b) and (c)). Enough abrasives can provide an excellent polishing performance of prepared MAPs.
In addition, the morphology of CBN particle before and after was analyzed. Based on the analysis results of Fig. 3 (a) and (b), the original morphology of CBN particle is maintained. The sharp cutting edge of CBN particle are not destroyed after the atomization process. This is because the predetermined temperature of the alloy matrix is lower than the melting point of CBN (3000℃). Additionally, the rapid cooling rate of atomization process makes the liquid powder solidify in a short time. Therefore, the high temperature around CBN particles have little effect on the damage to the cutting edges. The CBN particle maintain its original finishing capability.”
- In "5.2. Interaction effects of process parameters", the analysis and interpretation of experimental results in Figure 7 and Figure 8 are lacking.
Response:The analysis and interpretation of experimental results in Figure 7 and Figure 8 was added in our revised manuscript. The modified content is as follows:
“For higher working gap, the magnetic lines of forces become loose and the strength of formed abrasive tool is weak. The available magnetic force of MAPs decreases causing CBN particles insufficient indentation depth on cemented carbide. The original burrs and scratches still remain on the workpiece surface which results a low Conversely, at a low working gap, the large magnetic flux density can provide a high indentation of CBN particles to remove sufficient material from the surface. Moreover, the value of increases first with the increasing rotational speed, the decrease appears after a certain rotational speed. The rotational speed determines the collision number between CBN particles and material surface coupled with the stability of abrasive tool. The small produced at relatively slow rotational speed is caused by the insufficient collision.”

Reviewer 3 Report
Dear Authors,
I am not convinced with technical details of work and my concerns are listed below:
11. What is the percentage improvement in surface finish achieved using MAF and Integrated MAF processes by other researchers. Mention the results in reported literature.
22. Literature related to MAF process is outdated and Literature from year 2019 to 2022 is totally missing in the present manuscript.
33. Relevant references are totally missing. Strictly use references related to MAF topic such as chemically assisted MAF of Inconel 625 tubes. https://doi.org/10.3390/mi13081168 so on.
44. Highlight the novelty and scope of present work in manuscript.
55. How was the gap between tool and work piece measured?
66. How the range of parameters was decided. Explain?
77. Why ? the central composite design (CCD) has been chosen to design the experiments
88. How much quantity of CBN abrasives was used in gap during finishing and on which basis, the quantity was decided. Mention in manuscript.
99. Which types of magnets were used in the magnetic tool. Mention the complete tool details in the manuscript.
110. How is the groove opening can increase magnetic flux density? Explain and add in manuscript.
111. Is the magnetic flux density 0.338 T sufficient for experimentation. Have you done any magnetic shielding?
112. The manufacturer details of SAE 5W-30 oil lubricant is missing.
113. Reference style should be uniform and as per journal requirement.

Author Response
Ref.No.:Micromachines -2066800
Title: Polishing characteristics of cemented carbide using cubic boron nitride magnetic abrasive powders
Journal: Micromachines
We would like to express our sincere thanks to the reviewer for the constructive and positive comments.
Detailed response to Reviewer(s)' Comments:
- What is the percentage improvement in surface finish achieved using MAF and Integrated MAF processes by other researchers. Mention the results in reported literature.
Response:
Thanks for your advice. A new section “5.5 Comparison of polishing performance of MAPs” was added in our revised manuscript. The content is a brief of literature survey on MAF results that were focused on polishing different materials by the MAPs prepared by different processes. The percentage improvement in surface finish was mentioned. The added section is as follow:
Table 8 is the polishing performance of the MAPs prepared by different technologies. It can be seen that the MAF process can produce an excellent surface quality on different material. However, the present work prepared CBN MAPs can achieve a nanometer roughness on cemented carbide. The hardness of the workpiece seems to be the highest among the MAPs referenced. The high wear resistance and hardness of CBN enhance the finishing capacity of MAPs.
Table 8. Performance survey of MAPs.
|
Preparation technology |
MAPs |
Workpiece material |
Original Ra |
Finished Ra |
%Ra |
|
Mixing |
Fe/Al2O3 |
Ti-6Al-4V |
1.121μm |
0.046μm |
95.9% |
|
Sintering |
Fe/Al2O3 |
Steel718 |
2.8μm |
0.39μm |
86% |
|
Gel |
Fe/SiC |
SKD11 |
0.65μm |
0.110μm |
83% |
|
Plasma sprayed |
Fe/Al2O3 |
SS316 |
0.299μm |
0.068μm |
77% |
|
Alloy-Hardening |
CI-MAPs |
Zr-alloy |
0.361μm |
0.085μm |
76.5% |
|
Present work |
Fe/CBN |
cemented carbide |
0.3μm |
0.0576μm |
80.8% |
- Literature related to MAF process is outdated and Literature from year 2019 to 2022 is totally missing in the present manuscript.
Response:Thanks for your advice. We have read some references from the last 4 years (2019-2022), and in our revised manuscript those references have been cited:
[3] Fan ZH, Tian YB, Zhou Q, Shi Ch. Enhanced magnetic abrasive finishing of Ti–6Al–4V using shear thickening fluids additives. Precis Eng 2020;64:300-306.
[4] Guo C, Zhang DL, Li XH, Liu J, Li F. A permanent magnet tool in ultrasonic assisted magnetic abrasive finishing for 30CrMnSi grooves part. Precis Eng 2021;72: 417–425.
[8] Qian C, Fan Z, Tian Y, Liu Y, Han J, Wang J. A review on magnetic abrasive finishing. Int J Adv Manuf Technol 2021;112: 619–634.
3. Relevant references are totally missing. Strictly use references related to MAF topic such as chemically assisted MAF of Inconel 625 tubes. https://doi.org/10.3390/mi13081168 so on.Response:We are very sorry for losing the relevant references in our original manuscript. It will be better if the recommended references mentioned were cited in our manuscript. Surely, they have done lots of work on study of the MAF process. Those references have been added in the revised manuscript.
[5] Singh G, Kumar H, Kansal HK, Sharma K, Kumar R, Chohan JS, Singh S, Sharma S, Li ChH, Krolczyk G, Krolczyk J. Multiobjective optimization of chemically assisted magnetic abrasive finishing (MAF) on Inconel 625 tubes using genetic algorithm: modeling and microstructural analysis. Micromachines 2022;13:1168.
[24] Liu GX, Zhao YG, Meng JB, Gao YW, Song Zh, Cao Ch. Preparation of AL2O3 magnetic abrasives by combining plasma molten metal powder with sprayed abrasive powder. Ceram Int 2022;48:21571-21578.
[25] Li WSh, Li JJ, Cheng B, Zhang XJ, Song Q, Wang Y, Zhang T, Seniuts U, Belostrkovsky M. Achieving in-situ alloy-hardening core-shell structured carbonyl iron powders for magnetic abrasive finishing. Mater Design 2021;212:110198.
4. Highlight the novelty and scope of present work in manuscript.Response:
Thanks a lot for your valuable review. The novelty of this study is to demonstrate the excellent polishing performance of MAPs prepared by atomization on superhard materials. Although the prepared Al2O3 MAPs have outstanding finishing and service life characteristics, the weak wear resistance and hardness of abrasive phase make produced MAPs perform poorly in terms of material removal efficiency on high hardness workpiece (ceramics, cemented carbide, etc). The original grinding marks and fault layers are still visible on the polished surface. Therefore, in this study, CBN micro particles with high wear resistance and hardness were used as abrasive phase to enhance the polishing performance of MAPs. According to the findings of this research, the atomized CBN MAPs can be an effective polishing tool for finishing the hard materials.
In our revised manuscript, the novelty of this study was stated. For example, the revised abstract is:
“…In order to improve the poor finishing performance and short service life of MAPs in polishing super-hard materials, a double-stage atomization technique was used to successfully manufacture MAPs with a CBN as abrasive phase. The prepared results show that CBN abrasives with their original structure were deeply and densely embedded on the surface of spherical MAPs. … According to MAF results, the strong cutting ability of atomized CBN MAPs improved the surface roughness of cemented carbide over 80% at the optimum parameters. The strong cutting ability of atomized CBN MAPs can produce good surface quality on the hard materials. The findings of this research can promote a large-scale application of MAF technology in the surface polishing of hard materials.”
The revised introduction is:
“The weak wear resistance and hardness of abrasive make developed MAPs perform poorly in the material removal efficiency. In order to improve the finishing performance of MAP, super-hard cubic boron nitride (CBN) fine particle was selected as abrasive. CBN abrasives and metal liquid were atomized by a double-stage atomization. MAF test was done to evaluate the polishing performance of atomized MAPs on cemented carbide.”
Other partial modifications have been marked in red in our revised manuscript.
- How was the gap between tool and work piece measured?
Response:The space between workpiece and magnetic pole is called working gap. No MAPs are adsorbed on the magnetic pole before MAF process. The CNC milling machine will adjust the working gap. After determine the working gap and then absorb the MAPs to the magnetic pole. The working gap is filled with the MAPs.
- How the range of parameters was decided. Explain?
Response:
The reason of the range of the each parameter was explained in the revised manuscript. The added content is as follow:
The range of the working gap was selected based on the magnetic field intensity. The surface roughness of cemented carbide polished under a greater 3mm working gap had a little change. The upper limit of working gap was selected 3mm. The rotational speed of magnetic pole will affect the stability of formed magnetic abrasive tool. Small rotational speed limit the finishing efficiency of MAF process while high rotational speed make the MAPs leave the working gap. Thus the range of rotational speed was 1000-2000rpm. Low feed rate would mean a high processing time of the workpiece while high feed rate would decrease the finishing effectiveness. For polishing hard material, the upper limit of feed rate was selected a relatively small value (5mm/min). The mesh number was selected from coarse to fine size of atomized MAPs. Table 3 shows the five levels of selected four factors in the experiment. The quality of MAPs can affect the stability and stiffness of the abrasive tool at a certain working gap. According to the value of working gap and the surface area of magnetic pole, the optimal weight of MAPs used at different working gaps was determined, as shown in Table 4. To reduce the friction heat, SAE 5W-30 oil lubricant provided by Shandong Hanneng Company was applied in MAF process. 0.2g oil lubricant was used in each orthogonal experiment.
- Why the central composite design (CCD) has been chosen to design the experiments?
Response:The aim of designing experiments is to establish a relation between the process parameters and the output. Central composite design methodology is commonly used techniques which involved small number of experiments and yields results with good accuracy. In addition, the central composite design can predict a second order behavior of the response for the process parameters. Therefore, the CCD technique has been selected for the present study to obtain a second order model.
- How much quantity of CBN abrasives was used in gap during finishing and on which basis, the quantity was decided. Mention in manuscript.
Response:
The quantity of MAPs is decided by the value of working gap and the surface area of magnetic pole. Therefore, the basis for quantity selection was added in the revised manuscript.
“Table 3 shows the five levels of selected four factors in the experiment. The quantity of MAPs can affect the stability and stiffness of the abrasive tool at a certain working gap. According to the value of working gap and the surface area of magnetic pole, the optimal weight of MAPs used at different working gaps was determined, as shown in Table 4.”
- Which types of magnets were used in the magnetic tool. Mention the complete tool details in the manuscript.
Response:
The details of the magnetic tool was added in our revised manuscript.
“a permanent magnet (Nd-Fe-B, BHmax/45 MGOe) was selected as a magnetic field generator”.
- How is the groove opening can increase magnetic flux density? Explain and add in manuscript.
Response:According to the formula H=F/A, the purpose of the grooves openings was to reduce the circumference area of the permanent magnet. The reduction of the circumference area (A) will increase the magnetic field intensity (H). The section “3.2. Magnetic tool design” was revised in the manuscript.
“Therefore, in the experiments, a permanent magnet (Nd-Fe-B, BHmax/45 MGOe) was selected as a magnetic field generator. Then the MAPs are arranged along the magnetic line of force and formed a cutting tool. The magnetic force (F) acting on the MAPs under a magnetic field is:
(1)
Where, d and are the size and magnetic susceptibility of MAPs,respectively. H and are the magnetic field intensity and magnetic field intensity gradient at the location of MAPs, respectively.
From Eq.3, the magnetic force of MAPs is proportionate to the magnetic field intensity of the magnetic field. In order to improve the magnetic field intensity, the permanent magnet was designed to conduct groove openings. The purpose of the grooves openings was to reduce the circumference area of the permanent magnet (H=F/A, where, F is the magnetic flux quantity, and A is the circumference area of magnet) [21].”
- Is the magnetic flux density 0.338 T sufficient for experimentation. Have you done any magnetic shielding?
Response:Thanks for your advice. The hysteresis curve of CBN MAPs is illustrated in Fig. A.It can be seen that the MAP can reach its saturation magnetization when the external magnetic field strength is about 3000 Oe. The magnetic pole can provide a 0.338T (3380 Oe) magnetic field strength after groove opening. The magnetic flux density is sufficient for the MAF process. In addition, we studied the effect of magnetic pole cover (copper material) to the change of external magnetic field. The cover can reduce the magnetic field intensity in the circumferential direction of the magnetic pole, and then reduce the adsorption of MAPs on the periphery of magnetic pole. In our research, the effect of magnetic pole with groove openings will also change the distribution of MAPs and improve the utilization rate of MAPs. Therefore, we did not add magnetic shielding.Fig.A Hysteresis loops of prepared CBN MAPs.
- The manufacturer details of SAE 5W-30 oil lubricant is missing.
Response:
The details of oil lubricant was added: To reduce the friction heat, SAE 5W-30 oil lubricant provided by Shandong Hanneng Company was applied in MAF process. 0.2g oil lubricant was used in each orthogonal experiment.
- Reference style should be uniform and as per journal requirement.
Response:Reference was checked again to make sure that the style was under the requirement of the journal.

Round 2
Reviewer 1 Report
The authors have fully addressed the issues raised by the reviewers and the updated version can be accepted for publication.
Reviewer 2 Report
The format of the reference is not standard, please modify it, for example, when more than three authors use et al.
Reviewer 3 Report
Dear Authors,
I appreciate your efforts for improving the manuscript as per recommendations.